# Parameters Affecting the Mechanical Properties of Three-Dimensional (3D) Printed Carbon Fiber-Reinforced Polylactide Composites

**DOI:** 10.3390/polym12112456

**Published:** 2020-10-23

**Authors:** Demei Lee, Guan-Yu Wu

**Affiliations:** Department of Mechanical Engineering, Chang Gung University, Taoyuan City 33302, Taiwan; recoverism0928@gmail.com

**Keywords:** 3D printing, processing parameters, tensile strength, impact strength, process optimization

## Abstract

Three-dimensional (3D) printing is a manufacturing technology which creates three-dimensional objects layer-by-layer or drop-by-drop with minimal material waste. Despite the fact that 3D printing is a versatile and adaptable process and has advantages in establishing complex and net-shaped structures over conventional manufacturing methods, the challenge remains in identifying the optimal parameters for the 3D printing process. This study investigated the influence of processing parameters on the mechanical properties of Fused Deposition Modelling (FDM)-printed carbon fiber-filled polylactide (CFR-PLA) composites by employing an orthogonal array model. After printing, the tensile and impact strengths of the printed composites were measured, and the effects of different parameters on these strengths were examined. The experimental results indicate that 3D-printed CFR-PLA showed a rougher surface morphology than virgin PLA. For the variables selected in this analysis, bed temperature was identified as the most influential parameter on the tensile strength of CFR-PLA-printed parts, while bed temperature and print orientation were the key parameters affecting the impact strengths of printed composites. The 45° orientation printed parts also showed superior mechanical strengths than the 90° printed parts.

## 1. Introduction

Polylactide (PLA) is extracted from natural sources and is decomposable and environmentally friendly. It has multipurpose applications in packing, pharmaceutical, textiles, automotive composites, and the biomedical and tissue engineering fields [1]. The material ranges from amorphous glassy polymer to semi-crystalline and highly crystalline polymer, with a glass transition temperature of 60–65 °C and a melting temperature of 173–178 °C. The molecular weight, crystallinity, and material geometry are factors which can be used to tailor the biodegradation time [2]. However, the applications of PLA are limited because of the less-than-optimum glass transition temperature, thermal dimensional stability, and mechanical ductility. The addition of carbon fibers to improve the final performance of materials has been a topic of interest.

Three-dimensional (3D) printing, also known as additive manufacturing [3], is a flexible process to produce parts of complex geometries. Three-dimensional printing has the advantages of rapid prototyping, mass customization, and coping with complexity over conventional manufacturing technologies. As a result, 3D printing has become one of the most promising technologies since the 1980s. The layer-by-layer creation process of 3D printing in producing products adds promising possibilities to conventional manufacturing processes [4]. Among the applications of 3D printing manufacturing, the fused deposition modelling (FDM) procedure [5], which extrudes the hot polymer melt of thermoplastics from a large coil and accumulates the materials on the growing work, has been the most popular one. The process moves the printing head along the X-axis and Y-axis of horizontal directions to form one layer at a time. Then, the printing head moves vertically in small sequential steps to create the next layer. A microchip in the 3D printing machine controls the position of the printing head to establish a three-dimensional product. Furthermore, to enhance the mechanical performance of the printed part, carbon fiber-reinforced filaments are used. By infusing carbon fibers into the base material, 3D-printed products can be stronger, stiffer, lighter, and more dimensionally stable.

Various research efforts have been completed on the 3D printing of PLA and PLA composites. Tian et al. [6] studied the recycling of 3D-printed continuous carbon fiber-filled PLA composites and the reuse of recycled raw material for the further 3D printing process. Wang et al. [7] studied the 3D printing of green composites, using filaments produced from PLA and kenaf fibers. Maloch et al. [8] proposed that the temperature of the extrusion nozzle and the layer thickness are two of the basic process parameters affecting 3D printing. Alafaghani et al. [5] investigated the independent effect of each processing parameter on the mechanical properties of FDM parts. Chen et al. [9] studied the tensile strength of a 3D-printed work piece and found that the part quality can be affected by the thickness of fill, fill rate, extruder speed, and extruder head temperature. Despite the number of these works, most of them focused on the printing of PLAs based on a time-consuming trial-and-error approach. Furthermore, the process optimization of the 3D printing of carbon fiber-reinforced (CFR)-PLA composites is limited.

In this study, an experimental approach with a fractional orthogonal array design [10] was conducted to investigate the effect of various processing factors on the mechanical properties of 3D-printed carbon fiber-reinforced (CFR)-PLA parts [11,12], so as to optimize the printing process. After printing, the mechanical properties of the printed CFR-PLA parts was measured by tensile and impact testers, while their morphology was characterized by a scanning electron microscope (SEM). In addition, a differential scanning calorimeter (DSC) was employed to examine the thermal characteristics of the printed CRF-PLA parts.

## 2. Materials and Methods

### 2.1. Materials

Short carbon fiber-reinforced polylactide (CRF-PLA) filaments (K Glory Industrial Co. Ltd., Taipei, Taiwan) possessing a diameter of 3.0 mm were employed for the 3D printing experiments. Virgin PLA filaments were also adopted as a comparison.

### 2.2. Experimental Setup

A commercially available FDM printer (V2-B Dual Extruder Printer, Kraftmaker, Taipei, Taiwan) was set up for the experiments. As shown in Figure 1A, it has a printing resolution of 200 μm.

Tensile test and impact test part geometries were selected for the parts. Figure 2 shows the arrangement and dimensions of the parts. A solid modelling computer-aided design (CAD) computer program (Solidworks, Waltham, MA, USA) was used to generate the code (Figure 1B) required for the 3D printing process.

After printing, following the ASTM D638 standard the tensile strength of the 3D-printed specimens was assessed by a Lloyd tensiometer (AMETEK, USA) equipped with a 2500 N load cell. The pulling of the clamps was set at a rate of 50 mm/min. Meanwhile, the impact strength was measured on a Charpy Ceast Resil 5.5 Impact Strength Machine (CEAST S.p.a., Torino, Italy), based on the ASTM D6110 standard. Measurements were made three times on each tensile and impact test part.

### 2.3. Experimental Parameters and Orthogonal Design

For the 3D printing of CFR-PLA composites, five processing variables, including print orientation, fill density, bed temperature, nozzle temperature, and print speed, were selected for the experimental analysis. The print orientation was either 45° or 90°. The fill density was set to be 40%, 50%, or 60%. The bed temperature was fixed at 60, 70, or 80 °C. The nozzle temperature was at 220, 230, or 240 °C. The printing nozzle speed was set to move at 50, 55, or 60 mm/s. Table 1 shows the selected factors with their associated levels.

With the selected factors and levels of the factors, a Taguchi L18 orthogonal array design [13] (Table 2) was created for the analysis. The array contains a minimized number of required experiments in determining all the parameters affecting the performance [13]. The optimal mechanical properties of a 3D-printed part can be obtained with the combination of all the conditions that result in the highest tensile and impact strengths.

### 2.4. Characterization of Printed Parts

A field emission scanning electronic microscope (JEOL Model JSM-7500F, Tokyo, Japan) was used to examine the morphology of the 3D-printed parts. The thermal properties of the CFR-PLA filaments of the printed parts was assessed with a TA Instruments model DSC25 differential scanning calorimeter (New Castle, DE, USA). The scan temperature was set to range from 30 to 250 °C, where the heating rate of the specimen was 10 °C/min. 

## 3. Results

Specimens for the tensile and impact strength measurements were prepared using a commercially available FDM printer. Figure 3 displays the fractured specimens after the tensile and impact tests, and Figure 4 illustrates the surface images of the printed parts. When compared to that of virgin PLA, due to the added fibers the composite materials are more likely to clog during extrusion and result in a rather rough strip surface. In addition, tiny pores could be seen on the extruded strips. Figure 5 shows the images of the fractured surfaces of the printed virgin PLA and CFR-PLA composites. While virgin PLA exhibited a rather sharp fracture surface, CFR-PLA showed a rough surface, with carbon fibers clearly seen.

Table 2 lists the tensile and impact properties of the printed CFR-PLA parts, completed based on the Taguchi approach. The properties obtained from each experimental trial were assessed statistically.

### 3.1. Process Optimization

The signal/noise, S/N, ratio was used to denote the changes in goal value in response to diverse noise conditions. A higher value for the S/N ratio represents better settings of the control variables that diminish the influence of the noise variables. The minimization of influence of noise variables on the experiments can be achieved by maximizing the S/N. Since the objective of this study is the maximization of the mechanical properties, an equation with the-greater-the-better feature was adopted for the analysis:(1)$$\frac{S}{N}=-10\log_{10}[(\frac{1}{n})\sum_{i=1}^{n}\left(\frac{1}{y_{i}^{2}}\right)],$$
where *y_i_* is the measured tensile strength for the *i*th test part and n represents the number of test parts. The largest S/N ratios indicate optimal factor levels that minimize the noise sensitivity.

The variation in the tensile strength of the printed CFR-PLA parts due to various factors was assessed in accordance with the Taguchi method. Figure 6 shows the main effect plots for the 3D-printed tensile test parts, while Figure 7 shows the S/N ratios. The S/N ratios in Figure 7A suggest that the optimized factor levels that generated the maximum tensile strength were estimated to be A1/B2/C2/D1/E2. These optimal factor levels indicate a print orientation of 45°, a fill density of 50%, a bed temperature of 70 °C, a nozzle temperature of 220 °C, and a print speed of 55 mm/s. The S/N ratios for the impact strength of the 3D-printed CFR-PLA parts was also calculated, and the results are shown in Figure 7B. The optimal factor levels that created parts with the highest impact strengths were estimated to be A1/B3/C2/D1/E2. These optimal factor levels indicate a print orientation of 45°, a fill density of 60%, a bed temperature of 70 °C, a nozzle temperature of 220 °C, and a print speed of 60 mm/s.

### 3.2. Confirmation Experiments

When the optimum combination of factor levels for the tensile strength of printed PLA is excluded from the array in Table 2, an additional approach is adopted to evaluate the part strengths in response to the optimal factor levels. Presuming that there is no interplay among the selected factors, the predicted S/N ratio for the optimal levels of tensile strengths, *η*_A1/B2/C2/D1/E2_, is
*η*_A1/B2/C2/D1/E2_ = *η*_m_ + (*η*_A1_ − *η*_m_) + (*η*_B2_ − *η*_m_) + (*η*_C2_ − *η*_m_) + (*η*_D1_ − *η*_m_) + (*η*_E2_ − *η*_m_)= *η*_A1_ + *η*_B2_ + *η*_C2_ + *η*_D1_ + *η*_E2_ − 4*η*_m_(2)
where *η*_m_ is the mean S/N ratio for the test trials summarized in Table 2, and *η*_FN_ denotes the S/N ratio for factor F and level N. Based on the equation, the assessed S/N ratio of the 3D-printed PLA parts for the optimum levels, A1/B2/C2/D1/E2, was 30.18 dB.

A confirmation test was conducted using the optimized factor levels. The optimal level for the solution extrusion-printed parts is A1/B2/C2/D1/E2. The average part strength thus acquired was 34.56 MPa, corresponding to an S/N ratio of 30.73 dB. This value is higher than the predicted value of 30.18 dB, and also higher than all 18 test trials using other processing conditions in Table 2. Therefore, the confirmation test result indicates that the optimized factor levels adequately maximized the tensile strengths of the 3D-printed parts.

Meanwhile, the assessed S/N ratios for the impact strengths of the printed CFR-PLA parts for the optimized levels, A1/B3/C2/D1/E2, was 21.77 dB. A test was conducted to verify the impact strength at the optimized factor levels. The test result indicates that the impact strength was 14.33 J at the optimized factor levels. Additionally, the S/N ratio for this confirmation experiment was 22.74 dB. This value is higher than the assessed value of 21.77 dB and those accomplished by other sets of processing conditions in Table 2. The experimental results together confirm that the adequate maximization of the impact strengths of the printed CFR-PLA composites was achieved at the optimum levels.

### 3.3. Influences of Processing Parameters

The S/N ratio indicates the level of robustness of a production process by assessing the relative influences of the control variables (signals) and those of the uncontrolled variables (noises) [10,13]. The influence of each processing parameter on the product quality can consequently change with the variation in S/N. From Figure 7A, the influence of each parameter on the tensile strength of the CFR-PLA parts was, from high to low, bed temperature (ΔS/N = 5.036 dB), print orientation (ΔS/N = 0.410 dB), print speed (ΔS/N = 0.397 dB), nozzle temperature (ΔS/N = 0.285 dB), and fill density (ΔS/N = 0.166 dB). Meanwhile, in line with the data presented in Figure 7B, the significance ranking of each parameter on the impact strength of the FDM-printed CFR-PLA composites was bed temperature (ΔS/N = 4.071 dB), print orientation (ΔS/N = 1.143 dB), fill density (ΔS/N = 0.746 dB), fill speed (ΔS/N = 0.578 dB), and nozzle temperature (ΔS/N = 0.223 dB). For the variables chosen in this analysis, bed temperature was identified as the most important parameter affecting the tensile strength of the CFR-PLA-printed parts, while bed temperature and print orientation were found to affect mostly the impact strengths of the printed composites.

### 3.4. Thermal Properties

The thermal properties of the CFR-PLA filaments and CFR-PLA composites printed with bed temperatures of 70 °C and 80 °C were characterized, and the result is shown in Figure 8. Based on the melting and crystallizing enthalpies, the measured crystallinities were 20.6%, 19.3%, and 23.8%, respectively, for the CFR-PLA filaments and the 70 °C- and 80 °C-printed CFR-PLA composites. Obviously, the 80 °C-printed composite exhibited a greater crystallinity than the 70 °C-printed composite. Meanwhile, due to the quenching effect of the print bed, 70 °C showed a lower crystallinity than the filaments.

## 4. Discussion

The mechanical properties are the objective measures for the 3D printing process optimization. Two factors may affect the strengths of the printed composite parts. The first factor is the healing of polymer strips at the interfaces via polymeric chain diffusion, which regains the molecular entanglement as well as the strengths. The second one pertains to the morphology of the printed parts. When the printed parts possess incompletely fused tiny pores and cracks due to less than optimum printing conditions, stress concentration may occur at these sites when subjected to external forces. The printed parts strengths decreased accordingly.

The experimental results in Figure 7 (factor level C2) showed that the CRF-PLA parts printed with a bed temperature of 70 °C exhibited the greatest tensile and impact strengths. The composite parts printed with a bed temperature of 80 °C show the lowest mechanical strengths. While the heated bed is designed to eliminate part warping, the control of the heated bed temperature at the right level is critical to the part quality. Benwood et al. [14] characterized the thermal behavior of the printed PLA samples and found that the bed temperature has a significant influence on achieving crystallinity, cold crystallization, and melting peak. Up to the bed temperature of 75 °C, the shape of the DSC curves remained similar, with a sharp melting peak at around 150 °C. Although the increased bed temperature may lead to an increase in the crystallinity and relevant mechanical properties, this effect may be lost due to the equally important issue regarding the weld formation of individual model layers. Figure 9 shows the images of parts printed with different bed temperatures. The 70 °C-printed parts exhibited a smooth surface morphology, and a better healing is observed. Meanwhile, the 80 °C-printed parts displayed a rougher surface and less healing among the printed strips. The un-healed gaps among the strips may lead to stress concentrations, when subjected to external forces. The printed parts thus exhibited an inferior part quality.

The experimental data in Figure 7A,B (factor level A1) suggest that one can obtain 3D-printed CFR-PLAs with higher mechanical strengths by adopting a 45° orientation. This might be because 3D printing builds parts that have inherently anisotropic properties. Casavola et al. [15] proposed that the 90° orientation printing is observed to make the printed sample more brittle. With 45° printing, the strips possess an orientation that can better resist the external tensile or impact loads (Figure 10). Furthermore, the orientation also had an impact on the print time. The 45° printing took a little longer to print a part, which in turn also influenced the printed part quality.

The molecular chain entanglement among various strips is influential on the mechanical properties of the printed products. During the printing process, the temperature of the polymeric melt decreases after it is extruded from the nozzle. Increasing the print speed maintains the temperature of the polymeric strips at a desired level for longer. A higher temperature allows a molecular chain to entangle at the interfaces and, as a result, gain in strength. However, a too-high print speed may lead to print imperfections and part failures.

Fill density is the amount of plastic used on the inside of the print. A higher fill density represents more plastic on the inside of the print, and therefore results in a stronger object, as suggested by the experimental data in Figure 7A,B (factor level B3).

The experimental data suggested that setting the nozzle temperature at the lowest level to print the parts resulted in the greatest mechanical strengths (factor level D1 in Figure 7A,B). A too-high temperature, when combined with a high shear stress as the composite is extruded from the nozzle, increased the risk of material degradation. The printed mechanical may deteriorate accordingly.

This study has designed and implemented an experimental approach that utilized the Taguchi technique to maximize the mechanical properties of 3D-printed CFR-PLA products. The same methodological approach can be further applied to optimize the printing of other materials. With the satisfactory research result, the research design and approach can also be generalized and applied to other 3D-printed products.

## 5. Conclusions

In this work, we have successfully investigated the influence of diverse processing parameters on the 3D printing of CFR-PLA parts, employing the Taguchi approach. The following are the conclusions obtained from the experimental results:(1)Among the variables in this analysis, the bed temperature was found to be the most important parameter affecting the tensile strength of the CFR-PLA printed parts, while the bed temperature and print orientation were the key parameters affecting the impact strengths of the printed composites.(2)The 45° orientation-printed parts showed superior mechanical strengths compared to the 90°-printed parts.(3)Higher fill density indicates a higher degree of compactness of plastic on the inside of the print, and as a result a stronger printed object.(4)The mechanical strengths of the printed parts decreased with the nozzle temperature. A too-high temperature, when combined with a high shear stress as the composite is extruded from the nozzle, increased the risk of material degradation and deteriorated the product quality.

## Figures and Tables

**Figure 1 polymers-12-02456-f001:**
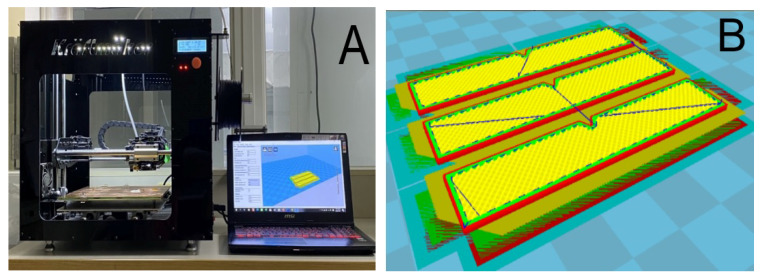
(**A**) Photo of the fused deposition modelling (FDM) printer; (**B**) the Cura interface used to control the printing process.

**Figure 2 polymers-12-02456-f002:**
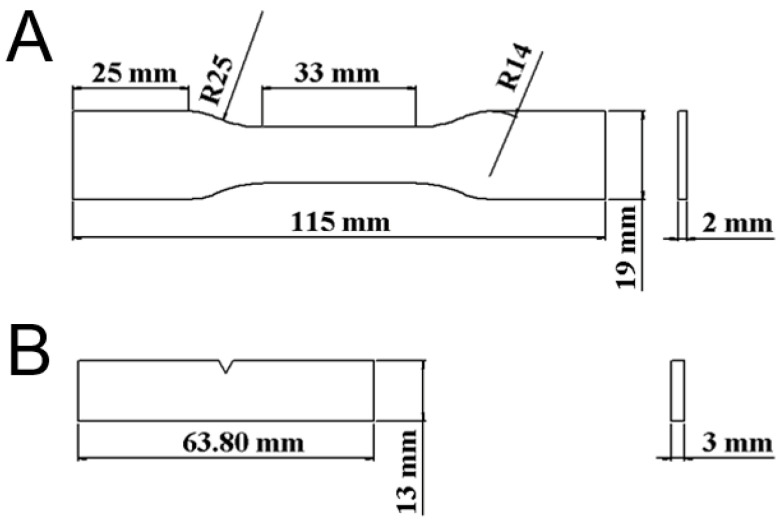
Layouts and dimensions of the (**A**) tensile test part and (**B**) impact test part.

**Figure 3 polymers-12-02456-f003:**
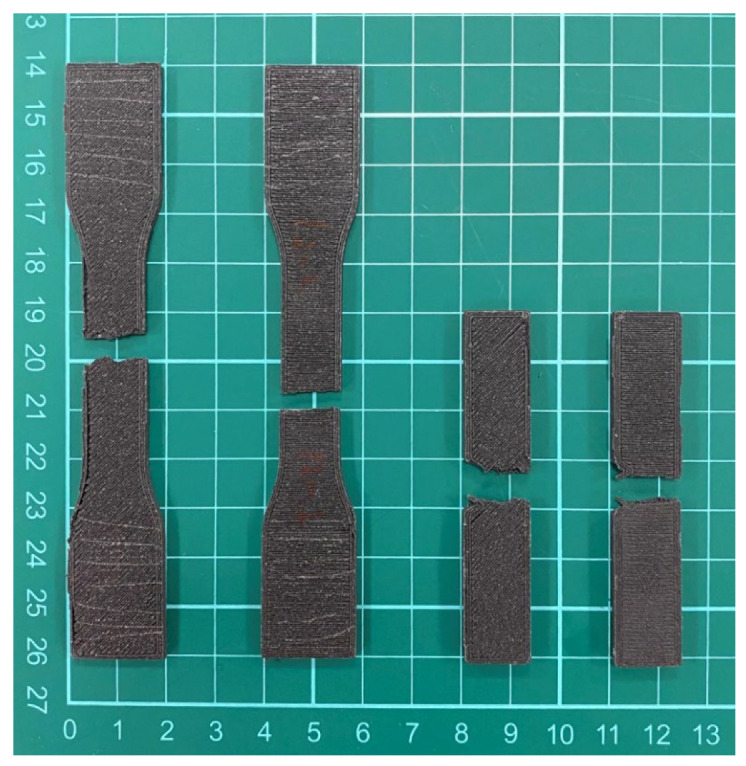
Photo of the fractured specimens.

**Figure 4 polymers-12-02456-f004:**
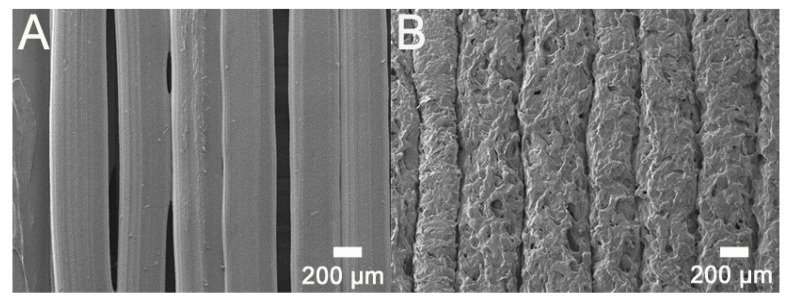
Surface images of 3D-printed (**A**) PLA and (**B**) CFR-PLA.

**Figure 5 polymers-12-02456-f005:**
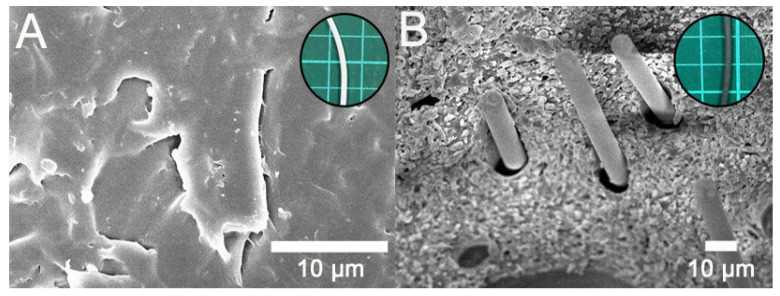
SEM images of fractured surfaces, (**A**) PLA, (**B**) CFR-PLA (upper corner shows the photo of the filament for 3D printing).

**Figure 6 polymers-12-02456-f006:**
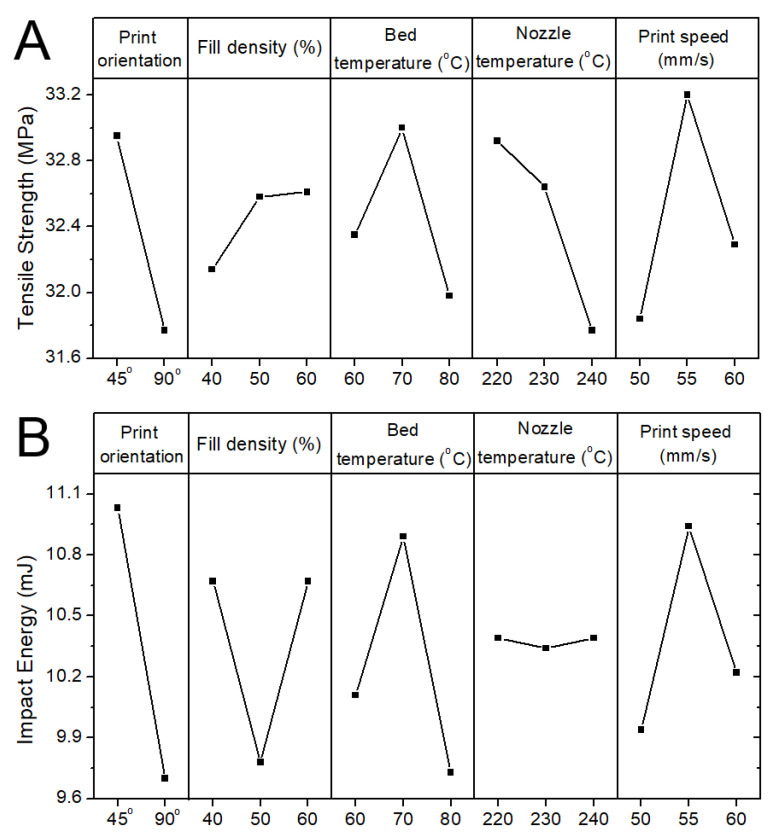
Main effect plots of the CFR-PLA parts for (**A**) tensile tests and (**B**) impact tests.

**Figure 7 polymers-12-02456-f007:**
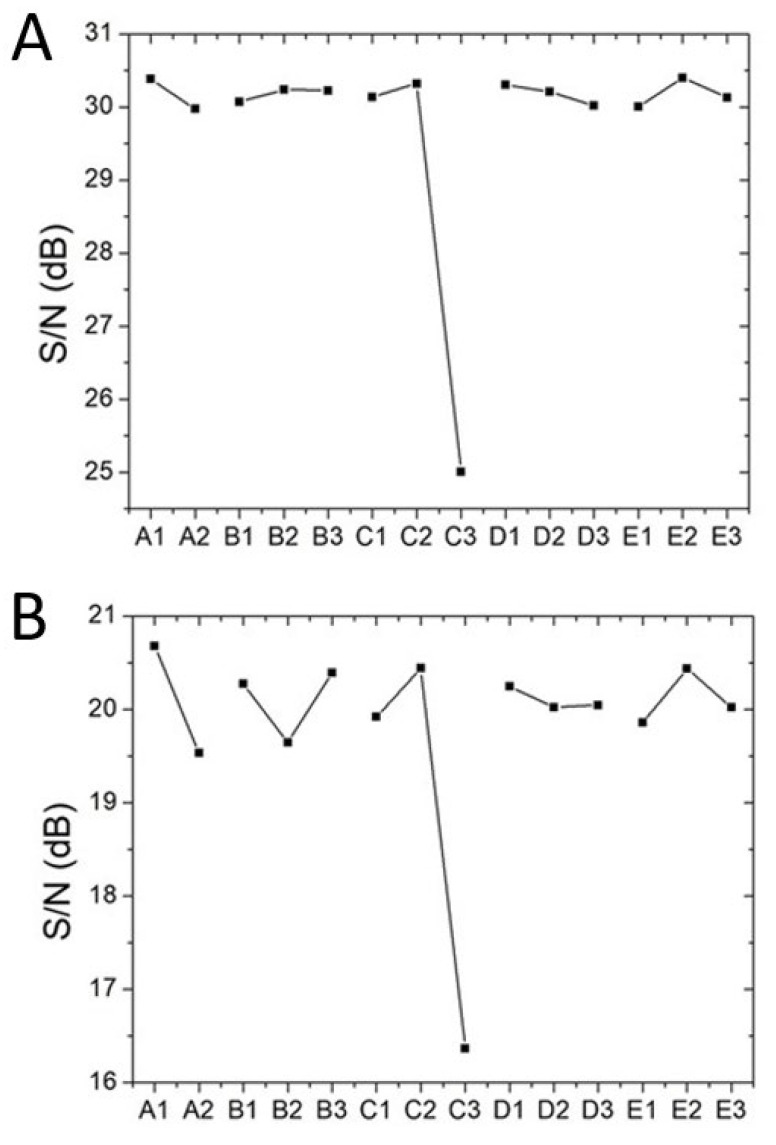
S/N ratios of the CFR-PLA parts for (**A**) tensile tests and (**B**) impact tests.

**Figure 8 polymers-12-02456-f008:**
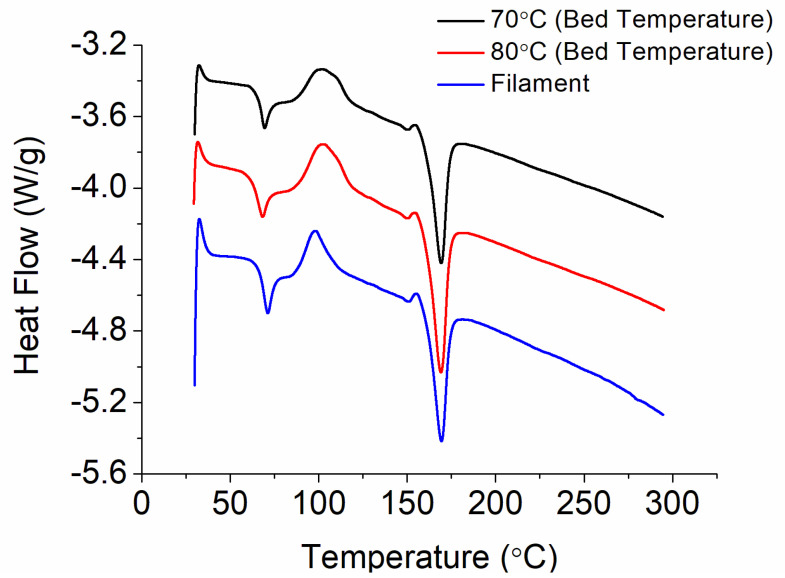
DSC curves of the CFR-PLA filament and CFR-PLA materials printed with different bed temperatures.

**Figure 9 polymers-12-02456-f009:**
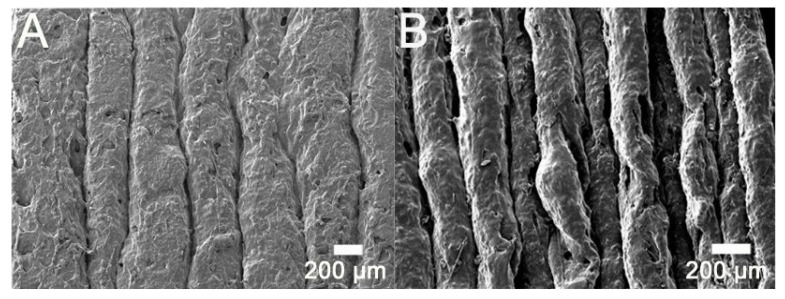
Surface images of the printed CFR-PLA with a bed temperature of (**A**) 70 °C and (**B**) 80 °C.

**Figure 10 polymers-12-02456-f010:**
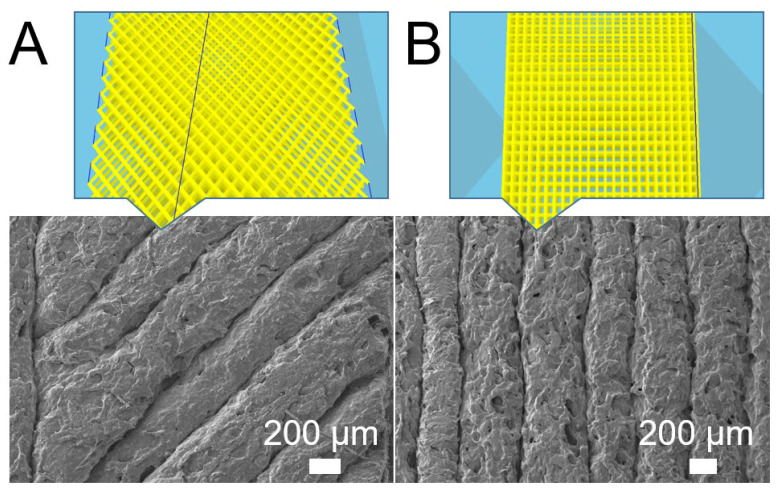
Print orientation and the SEM image of the printed composite parts ((**A**): 45°, (**B**): 90°).

**Table 1 polymers-12-02456-t001:** Parameters used for the 3D printing of carbon fiber-filled polylactide (CFR-PLA) composites.

Factor	A: Print Orientation	B: Fill Density (%)	C: Bed Temperature (°C)	D: Nozzle Temperature (°C)	E: Print Speed (mm/s)
**Level 1**	45°	40	60	220	50
**Level 2**	90°	50	70	230	55
**Level 3**	---	60	80	240	60

**Table 2 polymers-12-02456-t002:** L18 array used for the experiments.

Run	A	B	C	D	E	Tensile Strength (MPa)	S/N (dB)	Impact Strength (mJ)	S/N (dB)
**1**	1	1	1	1	1	33.29	30.38	10.33	20.26
**2**	1	1	2	2	2	34.05	30.55	14.00	22.40
**3**	1	1	3	3	3	31.23	29.88	12.00	21.52
**4**	1	2	1	1	2	33.31	30.44	10.33	20.26
**5**	1	2	2	2	3	33.54	30.51	9.67	19.67
**6**	1	2	3	3	1	32.31	30.19	9.33	19.37
**7**	1	3	1	2	1	33.73	30.48	10.67	20.53
**8**	1	3	2	3	2	33.51	30.50	11.33	20.76
**9**	1	3	3	1	3	33.65	30.52	11.67	21.32
**10**	2	1	1	3	3	29.39	29.31	8.33	18.23
**11**	2	1	2	1	1	31.10	29.75	10.33	20.26
**12**	2	1	3	2	2	33.76	30.56	9.00	18.98
**13**	2	2	1	2	3	32.08	30.03	9.67	19.47
**14**	2	2	2	3	1	31.93	30.08	10.00	19.64
**15**	2	2	3	1	2	32.33	30.18	9.67	19.47
**16**	2	3	1	3	2	32.27	30.17	11.33	20.76
**17**	2	3	2	1	3	33.84	30.54	10.00	19.91
**18**	2	3	3	2	1	28.70	29.1	9.01	19.08

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
