# Peer review of "Parameters Affecting the Mechanical Properties of Three-Dimensional (3D) Printed Carbon Fiber-Reinforced Polylactide Composites"

_polymers, 2020, doi:10.3390/polym12112456_

Round 1

Reviewer 1 Report

The manuscript deals with 3D printing of carbon fibre reinforced polylactide composites and assessment of effects of different factors such as print orientation, fill density, bed temperature, nozzle temperature and print speed. The manuscript is interesting and can be considered for publication in Polymers. However, a revision is needed to address the following comments/queries.

  1. The introduction does not establish the need for this with existing literature. This needs to be improved.
  2. Include ‘main effects plots’. Just S/N plots do not justify the statistical analysis!
  3. There are practically no significant changes in the mechanical properties / SN rations based on Table 2 (except Run 20)! Why? This leads to the following questions:
    1. How were these factors A, B, C, D and E were considered?
    2. How were the upper and lower limits picked up?
    3. Did authors do any preliminary experiments?
  4. Why is DSC incomplete? Why only these samples?
  5. Formatting:
    1. Plenty of grammatical errors. Significant efforts must be given to improve this.
    2. Inappropriate use of capital letters.

Summing up, I would recommend this manuscript for a major revision. It does not stand right to be considered for publication at all in its present form.

Author Response

Reviewer #1:

The manuscript deals with 3D printing of carbon fibre reinforced polylactide composites and assessment of effects of different factors such as print orientation, fill density, bed temperature, nozzle temperature and print speed. The manuscript is interesting and can be considered for publication in Polymers. However, a revision is needed to address the following comments/queries.

  1. The introduction does not establish the need for this with existing literature. This needs to be improved.

Response: The manuscript has been revised to better address the need for this current study (page 2).

  1. Include ‘main effects plots’. Just S/N plots do not justify the statistical analysis!

Response: The main effect plots have been provided in Figure 6.

  1. There are practically no significant changes in the mechanical properties / SN rations based on Table 2 (except Run 20)! Why? This leads to the following questions:
    1. How were these factors A, B, C, D and E were considered?
    2. How were the upper and lower limits picked up?
    3. Did authors do any preliminary experiments?

Response: Table 2 lists the 18 test runs in the analysis (there is no Run 20 in Table 2 !?). The results in Table 2 show that the highest impact strength (14.0 mJ in Run 2) is 55% greater than the lowest strength (9.0 mJ in Run 18). Significant changes were found in the experimental works, which testify the appropriateness of the Taguchi approach and the parameters selected in this study.

  1. Why is DSC incomplete? Why only these samples?

Response: The experimental results in this study suggest that the bed temperature is the key factor affecting the impact strengths of printed parts. This study thus examined the influence of bed temperature on the crystallinity of printed PLAs. The thermal properties of CFR-PLA filaments, and CFR-PLA composites printed with a bed temperature of 70°C and 80°C were characterized and the result is shown in Figure 8.The manuscript has been revised to better explain this (page 8).

  1. Formatting:
    1. Plenty of grammatical errors. Significant efforts must be given to improve this.
    2. Inappropriate use of capital letters.

Response: The manuscript has been thoroughly checked and revised to improve its readability.

Summing up, I would recommend this manuscript for a major revision. It does not stand right to be considered for publication at all in its present form.

Reviewer #2:

The manuscript investigated the paramaters for 3D printing carbon fiber-reinforced PLA composites according to orthogonal experimental design. The experimental design is suitable and the manuscript is clearly presented. I suggest it can be published after a minor revision.

  1. Ln 58-62: The introduction on other materials are not suitable here. I suggest to delete it.

Response: These lines have been removed.

  1. Ln 67-69: These sentence (This approach...fewer test trials.) are suggested to delete.

Response: These lines have been removed.

  1. Ln 207: The calculation on crystallinity should be based on the equation from melting and crystalling enthalpies.

Response: Based on the melting and crystallizing enthalpies, the measured crystallinities were 20.6%, 19.3%, and 23.8%, respectively, for CFR-PLA filaments, and 70°C and 80°C printed CFR-PLA composites. The manuscript has been revised to better address the experimental results (page 8).

  1. Please check the typing mistakes.

Response: The manuscript has been thoroughly checked to revise the typos.

Reviewer 2 Report

The manuscript investigated the paramaters for 3D printing carbon fiber-reinforced PLA composites according to orthogonal experimental design. The experimental design is suitable and the manuscript is clearly presented. I suggest it can be published after a minor revision. 1. Ln 58-62: The introduction on other materials are not suitable here. I suggest to delete it. 2. Ln 67-69: These sentence (This approach...fewer test trials.) are suggested to delete. 3. Ln 207: The calculation on crystallinity should be based on the equation from melting and crystalling enthalpies. 4. Please check the typing mistakes.

Author Response

(The authors gave the same response as above.)
